# Lower CSF Amyloid-Beta_1–42_ Predicts a Higher Mortality Rate in Frontotemporal Dementia

**DOI:** 10.3390/diagnostics9040162

**Published:** 2019-10-25

**Authors:** Daniela Vieira, João Durães, Inês Baldeiras, Beatriz Santiago, Diana Duro, Marisa Lima, Maria João Leitão, Miguel Tábuas-Pereira, Isabel Santana

**Affiliations:** 1Neurology Department, Centro Hospitalar e Universitário de Coimbra, 3000-045 Coimbra, Portugal; danielacgvieira@gmail.com (D.V.); duraes.jlo@gmail.com (J.D.); ines.baldeiras@sapo.pt (I.B.); hbmcsantiago@hotmail.com (B.S.); diana.duro@gmail.com (D.D.); marisalima5@hotmail.com (M.L.); jajao86@gmail.com (M.J.L.); isabeljsantana@gmail.com (I.S.); 2Faculty of Medicine, University of Coimbra, 3000-070 Coimbra, Portugal; 3Center for Neuroscience and Cell Biology, University of Coimbra, 3000-070 Coimbra, Portugal

**Keywords:** frontotemporal dementia, amyloid, cerebrospinal fluid, mortality

## Abstract

Frontotemporal lobar degeneration, the neuropathological substrate of frontotemporal dementia (FTD), is characterized by the deposition of protein aggregates, including tau. Evidence has shown concomitant amyloid pathology in some of these patients, which seems to contribute to a more aggressive disease. Our aim was to evaluate cerebrospinal fluid (CSF) amyloid-beta as a predictor of the mortality of FTD patients. We included 99 patients diagnosed with FTD—both behavioral and language variants—with no associated motor neuron disease, from whom a CSF sample was collected. These patients were followed prospectively in our center, and demographic and clinical data were obtained. The survival analysis was carried through a Cox regression model. Patients who died during follow up had a significantly lower CSF amyloid-beta_1–42_ than those who did not. The survival analysis demonstrated that an increased death rate was associated with a lower CSF amyloid-beta_1–42_ (HR = 0.999, 95% CI = [0.997, 1.000], *p* = 0.049). Neither demographic nor clinical variables, nor CSF total tau or p-tau were significantly associated with this endpoint. These results suggest that amyloid deposition in FTD patients may be associated with a higher mortality.

## 1. Introduction

Frontotemporal dementia (FTD) is a progressive neurodegenerative clinical syndrome, characterized by changes in behavior, executive function, and language. It is generally classified in three clinical variants: behavioral-variant FTD, non-fluent variant primary progressive aphasia, and semantic-variant primary progressive aphasia.

The neuropathological substrate, frontotemporal lobar degeneration, is characterized by neuronal loss, gliosis, and microvacuolar changes of the frontal lobes, anterior temporal lobes, anterior cingulate cortex, and insular cortex. Different subtypes are associated with abnormal deposition of different proteins, namely microtubule-associated protein tau, the TAR DNA-binding protein with molecular weight 43 kDa, or the fused-in-sarcoma protein [1].

On the contrary, amyloid-beta_1–42_ deposition is a hallmark of Alzheimer’s disease (AD). However, patients with FTD may have superimposed amyloid pathology [2]. Recently, amyloid-beta plaques have been found to facilitate the aggregation of tau [3] in AD models. Interestingly, in FTD patients with pathogenic mutations, amyloid has also been shown to be associated with a worse performance in several cognitive tests [4]. Additionally, CSF amyloid has been shown to be associated with higher volumetric loss in sporadic FTD patients [5].

We aimed to evaluate the value of cerebrospinal (CSF) amyloid-beta_1–42_ as a predictor of mortality in FTD patients.

## 2. Materials and Methods

### 2.1. Patient Selection

Our study included patients that fulfilled the current diagnostic criteria for behavioral variant of FTD [6] or language variants [7] with probable FTD-associated pathology, recruited at the Dementia Outpatient Clinic of the Centro Hospitalar e Universitário de Coimbra (Portugal). Defined inclusion criteria mandated a full neuropsychological evaluation and CSF biomarkers analysis. Patients who presented with motor neuron disorder at any point along the follow-up period were excluded. For the survival analysis, we considered the interval between the time of the lumbar puncture (LP)—baseline—and the time of death.

These patients are part of a prospectively evaluated cohort at our center. For the purpose of this study, we collected demographical data (age of onset, age at LP, gender, family history, and years of formal education), vascular risk factors at the time of the LP (history of diabetes, high blood pressure, dyslipidemia, atrial fibrillation, obesity, heart failure, coronary artery disease, heart valve disease, alcohol abuse, smoking, sleep obstructive apnea, and stroke), Mini-Mental State Examination (MMSE), and Clinical Dementia Rating (CDR) score at the time of LP.

The study was conducted according to the revised Declaration of Helsinki and Good Clinical Practice guidelines. Informed consent was given by all study participants or their legal next of kin. This project was approved by the ethics committee of our center, with the code HUC-43-09, prior to the beginning of the study.

### 2.2. CSF Biomarkers

CSF samples were collected as part of the routine clinical diagnostic protocol. Pre-analytical and analytical procedures were done in accordance with the Alzheimer’s Association guidelines for CSF biomarker determination [8]. CSF samples were collected in sterile polypropylene tubes, immediately centrifuged at 1800 g for 10 min at 4 °C, aliquoted into polypropylene tubes and stored at −80 °C until analysis. CSF Aβ42, T-tau, and P-tau were measured separately by commercially available sandwich ELISA kits (Innotest, Fujirebio, Belgium), as previously described [9,10]. External quality control of the assays was performed under the scope of the Alzheimer’s Association Quality Control Program for CSF Biomarkers [8].

### 2.3. Statistical Analysis

Statistical analysis was performed using the SPSS statistical software (version 20.0; SPSS Inc., Chicago, IL, USA). Categorical data are presented as frequency (percentage) and were compared using χ^2^-test. Ordinal or discrete risk factors are represented using median values and compared using the Mann−Whitney U test. Comparison of the measured variables between the different groups was performed using a Student’s t-test or γ^2^ test.

To assess for independent associations with death, a Cox regression was performed, adjusted for the collected data: age at onset, age at LP, CDR at the time of LP, CSF biomarkers, and behavioral versus language variants. Vascular risk factors with significant differences between patients who died during follow up and those who did not were also included. All the assumptions of these models were verified. Statistical significance was set at α = 0.05.

## 3. Results

We included 116 patients. Eight patients were lost to follow-up. Nine were excluded for having developed a motor neuron disorder. Finally, we included 99 patients, with a mean follow-up duration of 5.0 (±2.8) years. Mean age of onset was 61.4 (±9.3) years old. Male patients composed 46.9% of the cohort. Median education was 4.0 (IQR = 5.0) years. From the whole sample, 32 (33.3%) patients died during the follow-up period (average follow-up period for this subset of 5.0 ± 3.0 years). Table 1 presents the comparison of patients who died and those who did not during the follow-up period in terms of the studied variables. Vascular risk factors prevalence and comparison may be found in Appendix A.

In terms of vascular risk factors, patients who died during follow-up had a significant lower prevalence of high blood pressure (28.1% vs. 52.2%, *p* = 0.024), diabetes (6.3% vs. 23.9%, *p* = 0.033), and dyslipidemia (18.8% vs. 49.3%, *p* = *0*.004). No significant differences were found in the other vascular risk factors (Appendix A).

In the univariate analysis, death was associated with CSF amyloid-beta_1–42_ levels and CDR. In Cox regression (Table 2), the risk of death was associated with CSF amyloid-beta_1–42_ levels (HR = 0.999, 95%CI = [0.997, 1.000], *p* = 0.049).

## 4. Discussion

We report a relatively large size sample of thoroughly studied frontotemporal dementia patients. We found a significant independent association between lower CSF amyloid-beta_1–42_ levels and higher mortality.

There is a lack of good FTD diagnostic and prognostic biomarkers. Neurofilament light chain, total Tau [11], and phosphorylated-Tau/total-Tau ratio have been reported to have some clinical value [5,12,13,14,15]. Neurofilament light chain has also been shown to be correlated with disease severity, brain atrophy, annualized brain atrophy, and survival [16]. Additionally, some studies have shown that amyloid-positive patients perform significantly worse in cognitive tests [4]. Also, low baseline amyloid-beta_1–42_ has been associated with faster radiographic change in bvFTD [5].

In our sample, amyloid-beta_1–42_ has shown value as a marker of higher mortality rate (HR = 0.999, 95% CI = [0.997, 1.000], *p* = 0.049). If, on one hand, the clinical value of this association might be low, the biological link it represents may be of great importance in the understanding of the biological processes underlying this (and possibly other) conditions. In fact, it has been recently suggested that amyloid-beta plaques enhance tau-seeded pathologies by facilitating neuritic plaque tau aggregation [3]. This facilitation may happen through different mechanisms, such as the creation of a microenvironment that enhances the recruitment of tau into fibrils, but also by impairment of intracellular protein degradation [17].

This may be one explanation for these findings, but other hypotheses may be put forward, such as the occurrence of co-pathology (19.2% of the patients had reduced CSF amyloid-beta_1–42_). However, the rather young age of these patients makes it less likely that Alzheimer’s pathology may cause a great disease burden. On the other hand, it could be that, in the presence of concurring FTD-related pathology, a less advanced Alzheimer’s pathology could cause a more severe dysfunction. There is evidence that extracellular tau regulates the production of amyloid-beta, by mediating neuronal hyperactivity [18]. The amyloid-beta_1–42_ values could, therefore, be a reflection of a downstream impact of extracellular tau. In the absence of pathologic confirmation, misdiagnosis in a part of the sample is always a possibility. But these patients have been extensively studied and followed for several years. Moreover, one would expect that AD patients (who would have lower CSF amyloid-beta_1–42_ values) would progress slower than the FTD patients [19], however, this is an inverse relationship of what we found.

One important consideration is concerning vascular risk factors, as there were some differences in their prevalence on univariate analysis. In fact, the patients who did not die during the follow-up have a higher prevalence of some of these risk factors (diabetes, high blood pressure, and dyslipidemia) than those who have died. This is in line with some previous studies [20]. This may mean that the patients with a more rapid course have other strong factors driving the progression (possibly genetic), rather than a more multifactorial disease in the group of patients who did not die. These results could also mean that some of the patients in the group of patients who did not die during follow-up may have vascular dementia rather than frontotemporal dementia, leading to a more slowly progressive disease.

The main limitation of our work is its retrospective nature, although the data were prospectively collected, the hypothesis may be biased. Additionally, we do not have pathological data, and, since frontotemporal dementia may be associated with different pathological patterns and deposition of different proteins, the association with CSF amyloid-beta_1–42_ may be true only for one of these pathologies. However, the hard endpoint that we have measured (progression), and the moderate size of the sample, support the generalizability of these results.

## 5. Conclusions

Our results suggest that CSF amyloid-beta_1–42_ may modulate mortality in frontotemporal dementia patients.

## Figures and Tables

**Table 1 diagnostics-09-00162-t001:** Comparison of the two groups in terms of the studied variables.

Variable	Mean Values for the Whole Sample	Mean Values for Patients Who Died during Follow-Up	Mean Values For Patients Who Did Not Die during Follow-Up	*p*
Sex (% male)	46.9	50.0	45.3	0.664
Variant (% behavioral variant)	85.4	87.5	84.4	0.683
Family history (% positive)	30.2	28.1	31.3	0.753
Age of onset (years)	61.4 (±9.3)	61.1 (±9.8)	61.5 (±9.2)	0.831
Age at LP (years)	63.5 (±9.6)	62.9 (±10.1)	63.9 (9.6)	0.664
Education (years) (median, IQR)	4.0 (IQR = 5.0)	4.0 (IQR = 5.0)	4.0 (IQR = 5.0)	0.791
MMSE at LP	21.1 (±6.8)	20.0 (±7.1)	21.6 (±6.4)	0.271
CDR at LP (median, IQR)	1.0 (IQR = 1.0)	1.0 (IQR = 1.0)	1.0 (IQR = 0.0)	0.016
Follow-up (years)	5.0 (±2.8)	4.5 (±2.6)	5.0 (±3.0)	0.468
CSF amyloid-beta_1–42_ (pg/mL)	677.5 (±301.8)	532.7 (±306.0)	731.8 (±279.2)	0.002
CSF tau (pg/mL)	338.9 (±371.1)	277.6 (±166.5)	365.2 (±429.4)	0.286
CSF phosphorylated-tau (pg/mL)	41.4 (±38.4)	34.8 (±16.2)	44.7 (±44.4)	0.223

LP: lumbar puncture. MMSE: Mini-Mental State Examination, CDR: Clinical Dementia Rating, CSF: Cerebrospinal Fluid.

**Table 2 diagnostics-09-00162-t002:** Cox regression results of the variables associated with the death rate (γ^2^ = 18.799, df = 10, *p* = 0.043, −2 log likelihood = 225.540).

Variable	HR	95% CI	*p*
Age of onset	0.951	0.807, 1.120	0.546
Age at lumbar puncture	1.097	0.933, 1.289	0.261
Behavioral variant	0.634	0.147, 2.736	0.542
CDR	1.692	0.983, 2.914	0.058
CSF Amyloid-beta_1–42_	0.999	0.997, 1.000	0.049
CSF total tau	1.001	0.998, 1.004	0.514
CSF phosphorylated tau	0.985	0.956, 1.014	0.305
Diabetes	0.428	0.087, 2.118	0.298
High blood pressure	0.735	0.301. 1.794	0.499
Dyslipidemia	0.403	0.142, 1.147	0.089

HR: Hazard ratio; CI: Confidence interval; CSF: cerebrospinal fluid.

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
