# Peer review of "Lower CSF Amyloid-Beta1–42 Predicts a Higher Mortality Rate in Frontotemporal Dementia"

_diagnostics, 2019, doi:10.3390/diagnostics9040162_

Round 1
Reviewer 1 Report
The current form has well been revised. No comments for further improvements.
Author Response
Thank you for your previous comments and effort.
Nothing to add.
Reviewer 2 Report
A very interesting paper, and timely in that there is increasing awareness of the potential for mixed pathology across different dementia subtypes. It would be useful to add what the case mix was between bvFTD & language variants - in particular could any cases have been due to logopenic aphasia with primary Alzheimer's pathology?
Did the team have access to neuroimaging to assess degree of vascular burden or medial temporal lobe atrophy?
Minor language edits: first sentence of section 2.2 reads oddly, should be edited for clarity. Sentence at line 130 uses the word 'possibility' twice.
Author Response
A very interesting paper, and timely in that there is increasing awareness of the potential for mixed pathology across different dementia subtypes. It would be useful to add what the case mix was between bvFTD & language variants - in particular could any cases have been due to logopenic aphasia with primary Alzheimer's pathology?
Thank you for the comment. All the patients were classified by the latest widely accepted criteria and after extensive follow-up. However, we cannot exclude that, in fact, some patients (both those with language variants and those with behavioural variants) were caused, in fact, by Alzheimer pathology. However, as pointed out in the discussion if that is the case, that would, in theory, only reduce the power of the reported relationship, as Alzheimer patients would, on average, have a lower mortality rate. As reported in the table, 85.4% of the patients had a behavioural variant.
Did the team have access to neuroimaging to assess degree of vascular burden or medial temporal lobe atrophy?
Vascular burden and medial temporal lobe atrophy were not assessed in a systematic way that it could be included in the study. MRI, as amyloid-PET, was not performed in the whole sample, but in a great majority. Atrophy patterns and vascular burden were taken into account in the diagnostic process. The possibility if an impact of vascular lesions is minimized by taking into account all the vascular risk factors, which in the end had no association with mortality. Yet, we cannot completely exclude a possible effect of vascular burden in the association. Temporal lobe atrophy was a very rare finding in the sample.
Minor language edits: first sentence of section 2.2 reads oddly, should be edited for clarity. Sentence at line 130 uses the word 'possibility' twice.
Changed to: “CSF samples were collected as part of the routine clinical diagnostic protocol.”
This manuscript is a resubmission of an earlier submission. The following is a list of the peer review reports and author responses from that submission.
Round 1
Reviewer 1 Report
This study by Vieira et al. investigated whether CSF amyloid and tau biomarkers associated with following clinical outputs in around 100 patients with frontotemporal dementia. They concluded lower CF amyloid-beta 42 predicts a higher mortality rate in these patients. Their findings appear to be novel providing important insights into the disease pathogenesis. Methods and Discussion are well written.
Minor points
1. It is helpful to discuss the relationship between CDR and mortality, as they also reported significant associations between them.
2. In line 108, the hazard ratio and p-values are different from the reported values in Table 2.
Author Response
Please see the attachment
Reviewer 1
This study by Vieira et al. investigated whether CSF amyloid and tau biomarkers associated with following clinical outputs in around 100 patients with frontotemporal dementia. They concluded lower CF amyloid-beta 42 predicts a higher mortality rate in these patients. Their findings appear to be novel providing important insights into the disease pathogenesis. Methods and Discussion are well written.
Minor points
1. It is helpful to discuss the relationship between CDR and mortality, as they also reported significant associations between them.
We agree with the reviewer that this is missing from our paper.
Hence, we added a paragraph to the discussion:
Lines 128-131: “Higher CDR scores were also associated with a higher mortality risk. Despite the fact that CDR is usually considered to be an underachiever when classifying FTD patients[1,2], it is still a scale of disability; therefore, it is not unexpected that patients with a higher burden of disability (ie. in a more advanced stage of the disease) have a higher probability of dying in the near future.”
2. In line 108, the hazard ratio and p-values are different from the reported values in Table 2.
Changed..
Lines 108-109: In our sample, amyloid-beta1-42 has shown value as a marker of higher mortality rate (HR=0.999, 95%CI=[0.997, 1.000], p=0.022)”
1. Knopman, D.S.; Kramer, J.H.; Boeve, B.F.; Caselli, R.J.; Graff-Radford, N.R.; Mendez, M.F.; Miller, B.L.; Mercaldo, N. Development of methodology for conducting clinical trials in frontotemporal lobar degeneration. Brain : a journal of neurology 2008, 131, 2957-2968, doi:10.1093/brain/awn234.
2. Mioshi, E.; Hsieh, S.; Savage, S.; Hornberger, M.; Hodges, J.R. Clinical staging and disease progression in frontotemporal dementia. Neurology 2010, 74, 1591-1597, doi:10.1212/WNL.0b013e3181e04070.

Reviewer 2 Report
Dear Authors,
thanks for Your brief report entitled "lower CSF amyloid beta 1-42 predicts a higher mortality rate in frontotemporal dementia", submitted for publication.
Ninety-nine patients affected by FTD were retrospectively selected in order to evaluate cerebrospinal (CSF) amyloid beta 1-42 's values as a predictor of mortality.
The manuscript has serious flaws, and research was not correctly conducted.
The lack of data about comorbidities was a very significant flaw.
As You know, FTD patients do not die from FTD but from comorbidities, cardiovascular and infectious being the most frequent. The interference of comorbidities in amyloid beta 1-.42 levels depends on comorbidities and on their pharmacological approaches. No data about this point is present in your patient selection.
The retrospective natura of your report is another limit.
Frequently, when data from an incomplete database (such as in Your manuscript) are used, some conclusions may be statistically intriguing but without any clinical relevance. In Your study, the statistical significance of the association between amyloid beta 1-42 values and mortality rate was even modest (p-values = 0.022), when You performed a Cox regression (Table 2)
So, when I read your manuscript, I could not think that the evaluation of CSF amyloid beta 1-42's values - as performed in Your design study - may be considered a "good FTD prognostic biomarker", as You hypothesized in Your Discussion.
Finally, the possibility that some patients could have a frontal variant od AD ( please, read Ossenkoppele R et al, Brain 2015) had to be considered and thorough.
Author Response
Please see the attachment
Reviewer 2
Dear Authors,
thanks for Your brief report entitled "lower CSF amyloid beta 1-42 predicts a higher mortality rate in frontotemporal dementia", submitted for publication.
Ninety-nine patients affected by FTD were retrospectively selected in order to evaluate cerebrospinal (CSF) amyloid beta 1-42 's values as a predictor of mortality.
The manuscript has serious flaws, and research was not correctly conducted.
The lack of data about comorbidities was a very significant flaw.
As You know, FTD patients do not die from FTD but from comorbidities, cardiovascular and infectious being the most frequent. The interference of comorbidities in amyloid beta 1-.42 levels depends on comorbidities and on their pharmacological approaches. No data about this point is present in your patient selection.
We thank the reviewer for his/her comments, which raised important questions.
In fact, dementia per se is not the ultimate cause of death in frontotemporal dementia patients (according to one study, where the cause of death was ascertained by interviewing the patients’ caregivers[3]). However, probably, the dementia progression and associated disability probably concur for the frailty that increases the risk for other more acute comorbidities that ultimately lead to death (as the authors of that paper suggest, with major causes being pneumonia, choking on food, cardiovascular diseases and cachexia, the same as other dementias[4]). In that sense, in our opinion, the search for markers of more rapidly progressive diseases is of interest, and the use of time to death is a valid indicator, as it has been performed in other many studies.
In fact, we are reporting a group with an average age of onset of 61.4 years old and a mean follow-up (until death or last visit) of 5.0 years, with about a third dying during this follow-up. This means that, on average, there won’t be that many patients with meaningful cardiovascular risk factors and this alone would not (in our opinion) weight that much for the high rate of death, as the major risk factor for the ultimate cause of death is in fact the dementia itself. However, we have to accept that not accounting for the risk factors for cardiovascular and respiratory diseases is a limitation of the study.
Added:
Lines 133-137: “Additionally, not being controlled for other causes of death (or risk factors for), the interpretation of this data is limited. Most FTD patients die of respiratory problems, circulatory problems and cachexia[3]. However, these are the same that lead to death in other dementias[4], suggesting that dementia (or the disability caused by it) is the problem driving the risk for them.”
The retrospective natura of your report is another limit.
Frequently, when data from an incomplete database (such as in Your manuscript) are used, some conclusions may be statistically intriguing but without any clinical relevance. In Your study, the statistical significance of the association between amyloid beta 1-42 values and mortality rate was even modest (p-values = 0.022), when You performed a Cox regression (Table 2)
So, when I read your manuscript, I could not think that the evaluation of CSF amyloid beta 1-42's values - as performed in Your design study - may be considered a "good FTD prognostic biomarker", as You hypothesized in Your Discussion.
The reviewers raise an important issue here. In fact, the amyloid beta 1-42 values may have arguably no clinical interest (for now, at least). However, the finding of this association may signal an association with possibly a biological interest and this association may be of clinical interest in association with other markers. We think that this association merits dissemination and probably other studies looking at it in different ways.
We agree that the p-value is not that strong. However, it is there and after adjustment to a reasonably amount of variables. We think that the validity and generalizability of our findings are supported by the reports of findings with similar associations (ie. with a similar meaning) in other cohorts, as we have reported.
We are afraid we may have been misread somewhere along the paper, but we did not at any point claimed this to be "good FTD prognostic biomarker”.
Maybe in the conclusion, so we have changed it:
Lines 143-144: “Our results suggest a possible value of CSF amyloid-beta1-42 as predictor of mortality in Frontotemporal dementia patients.”
Finally, the possibility that some patients could have a frontal variant of AD ( please, read Ossenkoppele R et al, Brain 2015) had to be considered and thorough.
This is an important question.
The fact that there may be frontal variant AD patients in the sample is unquestionable. In fact, in the absence of pathological studies, the accuracy of the Rascovsky criteria is about 82%[5]. We additionally include CSF AD biomarkers and/or amyloid PET imaging in the evaluation of these patients, expecting to increase the specificity of the FTD label in our cohort. We cannot deny that there may be some patients with AD or even both pathologies in our cohort, but this is a limitation that most people in the field have to work with. However, as we stressed, the possible presence of AD patients in our sample would most probably reduce the likelihood of finding this association (ie. AD patients would most likely have lower amyloid-beta 1-42, and most likely a slower progression[6], therefore masquerading the association we found).
Lines 123-127: “In the absence of pathologic confirmation, the possibility of misdiagnosis in a part of the sample is always a possibility. But these patients have been extensively studied and followed for several years. Moreover, one would expect that AD patients (who would have lower CSF amyloid-beta1-42 values) would progress slower than FTD patients [6], that is, in an inverse relationship of that we have found.”
References
1. Knopman, D.S.; Kramer, J.H.; Boeve, B.F.; Caselli, R.J.; Graff-Radford, N.R.; Mendez, M.F.; Miller, B.L.; Mercaldo, N. Development of methodology for conducting clinical trials in frontotemporal lobar degeneration. Brain : a journal of neurology 2008, 131, 2957-2968, doi:10.1093/brain/awn234.
2. Mioshi, E.; Hsieh, S.; Savage, S.; Hornberger, M.; Hodges, J.R. Clinical staging and disease progression in frontotemporal dementia. Neurology 2010, 74, 1591-1597, doi:10.1212/WNL.0b013e3181e04070.
3. Nunnemann, S.; Last, D.; Schuster, T.; Forstl, H.; Kurz, A.; Diehl-Schmid, J. Survival in a German population with frontotemporal lobar degeneration. Neuroepidemiology 2011, 37, 160-165, doi:10.1159/000331485.
4. Koopmans, R.T.; van der Sterren, K.J.; van der Steen, J.T. The 'natural' endpoint of dementia: death from cachexia or dehydration following palliative care? International journal of geriatric psychiatry 2007, 22, 350-355, doi:10.1002/gps.1680.
5. Harris, J.M.; Gall, C.; Thompson, J.C.; Richardson, A.M.; Neary, D.; du Plessis, D.; Pal, P.; Mann, D.M.; Snowden, J.S.; Jones, M. Sensitivity and specificity of FTDC criteria for behavioral variant frontotemporal dementia. Neurology 2013, 80, 1881-1887, doi:10.1212/WNL.0b013e318292a342.
6. Kertesz, A. Rate of progression differs in frontotemporal dementia and Alzheimer disease. Neurology 2006, 66, 1607; author reply 1607, doi:10.1212/01.wnl.0000226826.42746.36.

Round 2
Reviewer 2 Report
Dear Authors,
Your reply seemed generic and not always true.
Firstly, I wanted to know the medical characteristics and data of Your patients. You replied referring data published by other investigators.....but what one may have observed may not apply to Your patients, and viceversa.
You reported a group with an average age of onset of 61.4 years. In this group, arterial hypertension, diabetes mellitus, chronic obstructive pulmonary disease... .may be present and give an important contribution to risk and mortality rates. Where these comorbidities present in Your patients ?
You wrote "additionally we included ....amyloid PET imaging in the evaluation of these patients....", but I did not found these data in Your manuscript.
In Your reply, some references (number 3 and number 4 ) were not appropriate.
In conclusion, I think that the quality of presentation and the significance of content of revised version of Your manuscript were low.